

# LakeSST: Lake Skin Surface Temperatures in French inland water bodies for 1999-2016 from Landsat archives

Jordi Prats[1], Nathalie Reynaud[1], Delphine Rebière[1, 3], Tiphaine Peroux[1], Thierry Tormos[2], Pierre-Alain Danis[2]

[1]UR RECOVER, Pôle AFB-Irstea hydroécologie plans d'eau, Irstea, F-13182, Aix-en-Provence, France
[2]Agence Française pour la Biodiversité, Pôle AFB-Irstea hydroécologie plans d'eau, F-13182 Aix-en-Provence, France
[3]CEREMA, Direction Centre-Est, F-63017 Clermont-Ferrand, France

*Correspondence to*: Jordi Prats (jordi.prats@irstea.fr, jordi.prats-rodriguez@gmail.com)

**Abstract.** The spatial and temporal coverage of the Landsat satellite imagery make it an ideal resource for studies on the long term evolution of lake surface temperature and for geographical studies of temperature patterns. The Lake Skin Surface Temperature (LakeSST) data set contains skin surface temperature data for 442 French water bodies (natural lakes, reservoirs, ponds, gravel pit lakes and quarry lakes) for the period 1999-2016 obtained from the thermal band of Landsat 5 and Landsat 7 archive images. The skin temperature measured by satellites differs slightly from water temperature in the first meters of the water column because of cool skin and warm layer effects. Nevertheless surface temperature parameterizations originally developed for the sea can be used to adjust LakeSST to commonly used lake water temperature, e.g. surface temperature or temperature of the first 1~2 m. Moreover, theoretically small differences are to be expected between the freshwater and seawater case for low wind speeds. In fact, at the reservoir of Bimont, the estimated cool skin effect was about -0.3 ℃ and -0.6 ℃ most of time, while the warm layer effect at 0.55 m was negligible in average, but could occasionally attain several degrees and a cool layer was often observed in the night. The overall accuracy of the satellite-derived temperature measurements was about 1.5 ℃, similar to other applications of satellite images to estimate freshwater surface temperatures. The LakeSST data are available at https://doi.org/10.5281/zenodo.1041746.



# 1 Introduction

Surface Water Temperature (SWT) is a key water quality parameter driving the ecological status of lakes (e.g. Shuter and Post, 1990; O'Reilly et al., 2003). Monitoring SWT is therefore an important issue especially in the context of climate changes. Nevertheless the SWT assessed over a national network of water quality monitoring are usually not frequent enough or limited to a restricted number of lakes. For example, in France, the implementation of the Water Framework European Directive implies the monitoring of 475 lakes but just 4 times a year only one year every three years, and such a frequency of observations is too low to detect long term trends. To improve the water quality monitoring and allow climate change effects survey, networks dedicated to monitor continuously temperature high frequencies (e.g. hourly time step) in the water column (1–4 m in vertical resolution) are used (Marcé et al., 2016), e.g. the Networking Lake Observatories in Europe (NETLAKE, https://www.dkit.ie/netlake) (Laas et al., 2016) or the Global Lake Observatory Network (GLEON, gleon.org) (Hamilton et al., 2015). However, such large scale networks mainly focus on large lakes and reservoirs while medium and small water bodies are more abundant than large water bodies (Verpoorter et al., 2014). In France, 15 relatively small water bodies (surface between 0.45 and 22 km²) are currently monitored. An additional source of data is research, but excepting rare cases, the collected data are disperse, often difficult to access and subject to homogenisation problems that may require long treatment times to incorporate them in a database.

Satellite thermal infrared (TIR) imagery, such as Landsat, can complete this monitoring effort by providing homogenous information on surface water temperature over large territories at moderate spatial and temporal scale. Moreover, remote sensing monitoring can collect information from inaccessible and non-instrumented water bodies, and long term and climate change studies can benefit from historical information starting in the 1980s (e.g. Schneider and Hook, 2010; Politi et al., 2012; Torbick et al., 2016). Because of its advantages, this technology has been increasingly applied to freshwater ecology in the last years (Dörnhöfer and Oppelt, 2016): satellite images have been used to study the water temperature of fluvial reaches (Lalot et al., 2015), reservoirs (Lamaro et al., 2013; Martí-Cardona et al., 2016), lakes (Marti-Cardona et al., 2008; Crosman and Horel, 2009) and crater lakes (Trunk and Bernard, 2008), as well as to monitor the thermal plumes from power plants (Zoran, 2011).

However, satellite-derived measurements correspond to the instant water temperature at the top of the surface (~10–20 μm deep), known as skin temperature (Donlon et al., 1999; Kawai and Wada, 2007). Skin temperatures can differ from SWT because the thermal structure of the first meters of the water column is not uniform under all conditions. At-sea measurements, of the skin surface temperature are usually lower than the temperature of the underlying water (cool skin effect), showing a dependence on wind speed (Fairall et al., 1996a; Minnett et al., 2011), and similar differences have been observed in freshwater environments (Cardenas et al., 2008). A warm near-surface layer can also appear on calm and sunny days, without necessarily excluding the presence of a cool skin (Donlon et al., 1999). Under such conditions the thermal gradient can attain several degrees in the upper few meters of the water column (Ward, 2006).

Nevertheless, satellite-derived water temperature data (skin temperatures) are relevant and sufficient to (i) demonstrate spatial and temporal patterns of surface water temperature for reservoirs and lakes (e.g. Schneider et al., 2009; Schneider and Hook, 2010; Prats and Danis, 2015); (ii) complement the data used for the calibration and validation of hydrodynamic and water quality models of lakes (Andréassian et al., 2012; Prats and Danis, 2017); and (iii) improve the estimation of surface

heat and gas fluxes (Lofgren and Zhu, 2000) that is important to understand the thermal behaviour of lakes (Henderson-Sellers, 1986).

This data paper presents the data set LakeSST of skin surface water temperature for 442 French lake water bodies derived from Landsat thermal infrared imagery. LakeSST was produced as part of a project funded by the French National Office for Water and Aquatic Environments (ONEMA) to assess the advantages and limitations of satellite data to monitor the water

temperature of inland water bodies. After recalling what is meant by surface temperature in this paper and the methodology used to derive skin temperatures from Landsat thermal infrared bands fully described by Simon et al. (2014), the precision and accuracy of the data set LakeSST are reported in view of surface temperature assessment. First, relations between skin temperature and surface temperature are dealt with in detail. Second the overall accuracy of LakeSST is assessed (i) by using water bodies located in overlapping areas of the images, and (ii) by confronting them to in-situ continuous temperature

measurements in five water bodies.

## 2 Study area and field data

### 2.1 Study area

The creation of this data set was funded by the ONEMA, now part of the French Agency for Biodiversity (AFB). The objective was to obtain long term measurements of French lakes which are regulatory monitored under the Water

Framework Directive. Initially this comprised a total of 475 lakes in metropolitan France of surface area larger than 0.5 km$^2$, including 64 natural lakes, 328 reservoirs, 43 ponds, 34 gravel pit lakes and 6 quarry lakes. However, the final data set contains data for 442 lakes, the rest having been excluded because of insufficient availability of data (Figure 1). The complete list of water bodies with geographical coordinates is included in the file 01_lake_data.txt of the data set. The geographical and morphometric data was extracted from the French lake data base (PLAN_DEAU) maintained by the AFB-

Irstea R&D Pole on Water Bodies Hydroecology at Aix-en-Provence (France).

### 2.2 In situ data for quality assessment

Satellite-based water temperature estimations were compared to continuous in situ measurements of water temperature at different depths at five French water bodies (Table 1, Figure 1). The in situ measurements were provided by the ONEMA and are part of a continuous lake temperature monitoring network. Water temperature was measured through chains of

HOBO WATER TEMP PRO V2/U22-001 thermistors installed on a buoy near the deepest part of the water body. The thermistors have a precision of ±0.2 ℃ between 0 ℃ and 50 ℃, and a resolution of 0.02 ℃ at 25 ℃. Temperatures were



measured at depths between 0.5 m and the bottom at intervals of 1 m to 4 m, depending on the depth (measurements were more spaced in the hypolimnion). Measurement frequency varied from once every 15 min to once every hour, depending on the water body and time period. In the analysis, a common 1-hour time step was used for all water bodies.

In one of the water bodies, the reservoir of Bimont (south-east of France, at 10 km from Aix-en-Provence), an additional
thermistor chain was installed, with the same type of thermistors dedicated to monitor in more detail the temperature gradient in the subsurface. The thermistors were placed at depths between 0.01 m and 0.55 m (at 0.01 m, 0.05 m, 0.10 m, 0.15 m, 0.20 m, 0.25 m, 0.35 m, 0.45 m, 0.55 m) between the 21 February 2014 and the 3 May 2016.

In addition to in situ data, meteorological data were gathered for the five French water bodies from the SAFRAN reanalysis data (Quintana-Seguí et al., 2008; Vidal et al., 2010), available for the study period at the nearest grid point at a daily
resolution. The following variables were extracted: air temperature, specific humidity, wind speed, liquid precipitation, solid precipitation, downwelling longwave radiation and incoming solar radiation. An adiabatic correction of -0.0065 ℃/m was applied to air temperature data to account for the difference of altitudes between the measurement point and the reservoir of Bimont. Additionally, we calculated solar elevation and extraterrestrial solar radiation following Lenoble (1993). We calculated the daily clearness index as
$$k_t = R_s/R_e \qquad\qquad\qquad (1)$$

where $R_s$ is the daily solar radiation at the Earth surface and $R_e$ is the extraterrestrial daily solar radiation through a plane parallel to the surface.

Finally, for the reservoir of Bimont, Météo-France kindly provided hourly meteorological data (air temperature, relative humidity, wind speed, solar radiation) measured at the meteorological station of Aix-en-Provence.

**3. Methods**

**3.1 Definitions of surface temperature**

As mentioned in the Introduction, the thermal structure of the first meters of the water column is not uniform. To avoid confusion, several definitions of surface temperature have been proposed for the marine environment, depending on the depth of measurement (Donlon et al., 2007; Kawai and Wada, 2007). We adapt (and adopt) these definitions for the
freshwater case:

- The interface surface temperature ($T_{int}$) is the theoretical temperature at the infinitesimally wide air-water interface.
- The skin surface temperature ($T_{skin}$) is the temperature of the first ~10–20 μm below the interface and it is dominated by heat conduction. It is measured with infrared radiometers.
- The subskin surface temperature ($T_{subskin}$) is the temperature of the first 1–1.5 mm of the water column. It is
measured with microwave radiometers.





- The temperature at depth $z$ ($T_z$), also known as (bulk) surface temperature, is the temperature measured at the said depth. It is the temperature measured by in situ sensors (thermistors, CTD, etc.). The French Agency for Biodiversity (AFB) proposes using $T_{0.5m}$ to monitor lake surface temperature (Rebière et al., 2014) and in the freshwater literature different reference depths are used for surface temperature, from 10−15 cm to several meters (e.g. Kettle et al., 2004; Toffolon et al., 2014).

- Finally, we will define the temperature of the surface mixed layer (SML) ($T_{sml}$) as the average temperature between the surface and the thermocline depth. The SML temperature $T_{sml}$ is calculated by integrating temperature measurements made with a thermistor chain or water quality profiler at different depths between the surface and the top of the metalimnion.

The bulk surface temperature can be related to the skin surface temperature through

$$T_{skin} = T_z + \Delta T_c + \Delta T_w \tag{2}$$

where $\Delta T_c$ is the cool skin effect and $\Delta T_w$ is the warm layer effect. Both effects are related to thermal inhomogeneities in the temperature distribution near the surface. The warm layer forms during the day, and is due to surface stratification caused by the absorption of solar radiation in the first meters of the water column (Fairall et al., 1996a). The magnitude of the warm

layer effect can attain several degrees (Fairall et al., 1996a; Kawai and Wada, 2007; Gentemann and Minnett, 2008). The cool skin effect is almost always present and arises from the cooling of the first millimetres of the water column because of the joint action of longwave radiation, sensible and latent heat fluxes (Fairall et al., 1996a). Its magnitude is of -0.1 ℃ to -0.5 ℃ (Fairall et al., 1996a; Donlon et al., 2002).

**3.2 Deriving skin temperature from Landsat imagery**

The Thermal InfraRed band both of the Landsat 5 TM (Thematic Mapper) and Landsat 7 ETM+ (Enhanced Thematic Mapper Plus) instruments were used for deriving skin temperature. The TIR band (band 6) on both sensors measures emitted radiation at wavelengths of 10.40–12.50 μm. Both satellites have similar orbits, with a delay of 8 days, and they fly over France at 10:00–10:30 UTC. Two consecutive scenes overlap at 5 % of their surface and are taken with a delay of 23.92 s (NASA, 2011). The TM instrument acquired Band 6 data at a 120-metre resolution and the ETM+ instrument acquired Band

6 data at a 60-metre resolution. TIR Landsat data were extracted from Landsat Climate Data Records (CDR) data. Landsat CDR consist of surface reflectance products from the Landsat archive (Masek et al., 2006) which are freely available from the United States Geological Survey (USGS). Thermal infrared data is provided as top-of-atmosphere (TOA) brightness temperature images resampled via cubic convolution to a pixel size of 30 m. Useful mask layers for clouds, cloud shadows, adjacent clouds, snow, land and water, and quality flags are also provided. The noise equivalent delta temperature of Landsat

measurements is 0.2–0.3 ℃ (Barsi et al., 2003).

TIR data must be corrected for emissivity and atmospheric effects if it is to be quantitatively useful (see Li et al., 2013). Emissivity variations for water-only pixels of relatively small and calm inland waterbodies, however, are insignificant in most practical applications. Atmospheric correction, on the other hand, is required to compensate for atmospheric absorption





and emission effects. Two possibilities are available for correcting single-band TIR images. The first is through the use of the radiative transfer equation (RTE) (e.g. Hook et al., 2004), in which transmissivity of the atmosphere, upwelling atmospheric radiance and downwelling atmospheric radiance are obtained through the use of radiative transfer modelling codes. This method, however, requires in situ radiosounding data obtained near the study area and near the acquisition time

of the image as input (Jiménez-Muñoz and Sobrino, 2003). The second possibility is to apply single-channel (SC) correction algorithms which are based on approximations of the RTE. Despite being less accurate, these algorithms crucially avoid dependence on in situ radiosounding data and are therefore better suited for satellite imagery archive studies {Sobrino, 2004 #2738}. Jiménez-Muñoz and Sobrino (2003), in particular, have developed an operational algorithm which relies solely on atmospheric water vapour content as ancillary data. This algorithm has been applied to several lakes and reservoirs (Lamaro

et al., 2013; Simon et al., 2014; Allan et al., 2016) and to the Guadalquivir River estuary (Díaz-Delgado et al., 2010). The version developed by Simon et al. (2014) was used for producing the LakeSST data set.

Total column water vapour data, required by the algorithm, was extracted from the ERA-Interim reanalysis data set (Dee et al., 2011) provided by the ECMWF (European Centre for Medium-Range Weather Forecasts) at a resolution of 0.25º. The water vapour content at the time the image was taken was interpolated from data at 6:00 and 12:00 UTC. During the

preprocessing phase, we extracted the valid pixels for each water body and image. For each image the preprocessing included the following steps:

1. Extraction of the TIR band, masks and metadata from the original hdf files.
2. Selection of water pixels using land-water mask and vector file containing the water body outline.
3. Suppression of unexploitable pixels by application of mask products associated to the image.

For the image processing, brightness temperature in the remaining pixels was converted to surface temperature following the single-channel algorithm proposed by Jiménez-Muñoz and Sobrino (2003), adapted to Landsat images (Jiménez-Muñoz et al., 2009). For further details on the algorithm or its calibration for Landsat thermal bands, please refer to these articles.

The algorithm is sensitive to variations of emissivity: a 1 % decrease in emissivity produced an increase in estimated temperature of 0.4~0.6 ℃. In this work the emissivity of water was assumed to be $\varepsilon = 0.9885$ (Lamaro et al., 2013).

However, emissivity can vary depending on factors such as the view angle and the concentration of suspended sediments. The emissivity of water varies with the zenith angle, mainly for angles above 40º (Masuda et al., 1988). The satellite Landsat 7 orbits the Earth at a height of 705 km, at a speed of 7.5 km s$^{-1}$ (NASA, 2011). At this speed, the variation in the viewing angle during the 23.92 s between two successive images is ~14º, so that there should not be an important effect on temperature linked to the variation of emissivity with the zenith angle. The emissivity of water can also be affected by the

concentration of suspended sediments, resulting in estimation errors of as much as 1 ℃ for a variation of emissivity of 0.01 (Wen-Yao et al., 1987). However, for freshwater the effect is only important for very high concentrations of suspended sediments ($\geq$ 10 g l$^{-1}$) (Wen-Yao et al., 1987).

The algorithm is applicable for a range of atmospheric water vapour content between 0.5 g cm$^{-2}$ and 2 g cm$^{-2}$; for vapour contents outside this range bias may be important (Jiménez-Muñoz et al., 2009). However, the algorithm is not very sensitive



to variations of atmospheric water vapour content: a 50% decrease in the value of $w$ produced a decrease in the estimated temperature of 0.2~0.3 °C.

A negative buffer was applied on the water bodies of each image to filter out pixels close to the bank and thus avoid potentially land-water mixed pixels. The width of the buffer was 170 m for Landsat 5 and 85 m for Landsat 7, which

approximately corresponds to the length of the pixel diagonal. All images for a given water body were then reprojected to a common Reference Coordinate System (WGS84, EPSG:4326) and resampled by cubic interpolation to a common 30 m grid. Images for which the water vapour content did not belong to the range of applicability of the algorithm (0.5–2 g cm$^{-2}$) were discarded. Images with negative values were also discarded (due to undetected ice cover, etc.). Summary statistics were calculated for each image: number of valid pixels, median temperature, mean temperature, standard deviation, minimum

temperature, maximum temperature, 25 % quartile, 75 % quartile.

### 3.3 Assessment of the relation between skin temperature and surface temperature

The applicability of surface temperatures measured by satellite to study the temperature of the surface mixed layer of lakes depends on the relation between surface temperature and subsurface temperatures. The skin temperature differs slightly from water temperature in the first meters of the water column because of cool skin and warm layer effects. Three methods were

implemented to assess these effects.

### 3.3.1 Warm layer estimations based on field measurements

We used data from the 0.55 m chain installed at the reservoir of Bimont to assess the relation between surface temperature and temperature at 0.55 m. The instrumentation used was not adapted to measure the cool skin effect, since the surface thermistor measured temperature at a depth of about 1 cm, below the cool skin. However, the Bimont data could be used to

measure the warm layer effect. We calculated the warm layer effect at 0.55 m as:

$$\Delta T_w(0.55m) = T_{0m} - T_{0.55m}. \tag{3}$$

### 3.3.2 Cool skin effect estimations depending on wind speed

Several parameterizations have been proposed to estimate $\Delta T_c$ as a function of wind speed measured at 10 meters above the ground surface $U_{10}$. Using data for the reservoir of Bimont, we applied the parameterizations obtained for the sea by Donlon

et al. (2002):

$$\Delta T_c = -0.14 - 0.30\exp(-0.27U_{10}), \tag{4}$$

Horrocks et al. (2003):

$$\Delta T_c = -0.11 - 0.35\exp(-0.28U_{10}), \tag{5}$$

Gentemann and Minnett (2008):

$$\Delta T_c = -0.13 - 0.22\exp(-0.350U_{10}), \tag{6}$$

and Minnett et al. (2011):



$$\Delta T_c = -0.130 - 0.724 \exp(-0.350 U_{10}). \tag{7}$$

### 3.3.3 Application of the COARE algorithm adapted to the freshwater environment

There are several models that take into account the cool skin and the warm layer effect at the sea (Price et al., 1986; Kantha and Clayson, 1994; Fairall et al., 1996a; Kawai and Kawamura, 2000; Gentemann et al., 2009). To analyse both the cool skin

and the warm layer effects for the case of the reservoir of Bimont, we used the model proposed by Fairall et al. (1996a), and implemented in the COARE bulk flux algorithm (Fairall et al., 1996b), that depends on the surface heat (shortwave irradiance, longwave irradiance, latent and sensible heat exchange) and moment fluxes (wind shear). We implemented the model in R based on the code of the COARE bulk flux algorithm version 3.0 (Fairall et al., 2003). We validated the implementation of the algorithm using four days of Moana Wave COARE test data (Figure 2). The FORTRAN code and test

data                    were                    obtained                    from
ftp://ftp1.esrl.noaa.gov/psd3/cruises/NTAS_2009/RHB/Scientific_analysis/programs/VOCALS2008_programs_leg1/coare/bulkalg/cor3_0/. In the original parameterization, the physical properties of water (viscosity, density, thermal and salinity expansivities and thermal conductivity) are constant and adapted to the seawater case. We modified the algorithm to calculate the physical properties of water as a function of temperature and salinity using expressions provided by Sharqawy

et al. (2010). An R file (Coare_3_0.R) with the implementation of the model is supplied as supplementary information. The simulation of $\Delta T_w$ for the test data was almost equal for both algorithms. However, the algorithm with physical properties of water depending on temperature and salinity simulated a less intense cool skin effect (with an average difference of 0.03 ℃). The difference was due to a too high value of the kinematic viscosity of $10^{-6}$ m$^2$ s$^{-1}$ in the original COARE algorithm. According to the tables published by Chen et al. (1973), the viscosity of seawater is $0.84 \cdot 10^{-6}$ m$^2$ s$^{-1}$ at 30 ℃ and $1.0 \cdot 10^{-6}$ m$^2$

s$^{-1}$ at 20 ℃. The average temperature of the test data at a depth of 6 m was 29.4 ℃, and the average value of kinematic viscosity of water in the modified version of the algorithm was $0.85 \cdot 10^{-6}$ m$^2$ s$^{-1}$.

We applied the modified COARE algorithm to the data of the reservoir of Bimont, assuming a salinity of 0 g kg$^{-1}$ and a maximum depth of the warm layer of 5 m (approximately the seasonal thermocline depth). We calculated the cool skin and warm layer effects also for the seawater case, assuming a salinity of 35 g kg$^{-1}$, using the same data, since it is of interest to

know if other algorithms applied at the side are applicable to the freshwater case and with which limitations. The use of salinity dependent functions for the physical properties of water, allowed us testing whether differences in the behaviour in the surface temperature between freshwaters and seawaters were theoretically expected because of the difference of salinity.

### 3.4 Assessment of the precision and accuracy of satellite-based surface water temperature measurements

Consecutive satellite scenes overlap by 5%. As a consequence, at these overlapping zones repeated temperature

measurements are available with just a few seconds of difference. We used the data from 45 water bodies located in the overlapping zones between consecutive scenes to estimate the precision of satellite-based surface water temperatures.



The assessment of the accuracy of satellite measurements was made by comparing them to in situ SML temperature and temperature at 0.5 m for 5 lakes with continuous monitoring (Table 1). We calculated the bias for the five water bodies with continuous measurements as the difference between field measurements and satellite temperature measurements at the nearest pixel ($T_{sat}$) as

$$b_{0.50m} = T_{sat} - T_{0.50m} \tag{8}$$

$$b_{sml} = T_{sat} - T_{sml} \tag{9}$$

where $b_{0.50m}$ is the bias in relation to temperature at 0.50 m and $b_{sml}$ is bias in relation to the SML temperature. The bias depended on the distance of the measurement location to the nearest pixel, because of spatial variability of the surface temperature (Figure 3). Absolute bias increased for distances greater than 500 m (the slope of the linear model of bias as a function of distance was not statistically significant at the 0.05 level, but we found $p$-value = 0.10). Most of the satellite-field data pairs separated by more than 500 m correspond to SCR04, the largest water body. Satellite temperatures measured near the field measurement point at this reservoir (near the dam) were warmer than in the open waters. To exclude this distance effect, we kept for further analyses the data pairs separated by less than 400 m.

### 3.5 Software used

We analysed the data using Python 2.7 and R 3.2.0 (R Core Team, 2015). We used the Python packages NumPy (van der Walt et al., 2011), Matplotlib (Hunter, 2007) and Pandas (McKinney, 2010) and the R package MASS (Venables and Ripley, 2002) and rLakeAnalyzer (Winslow et al., 2016).

## 4. Cool skin and warm layer effects

### 4.1 Theoretical salinity effects

The application of the COARE model revealed differences in the cool skin effect caused by the difference in salinity between the freshwater and seawater cases (Figure 4 a, c). For low wind speeds (approx. < 4 m s$^{-1}$) the skin effect was more important (by 0.03 ℃ in average) and the cool skin was thicker (by 0.2 mm in average) in the freshwater simulation. The difference could attain 0.2 ℃ for very low wind speeds. Above 4 m s$^{-1}$, the two cases show a very similar behaviour, with average differences in cool skin effect and thickness of 0.002 ℃ and 0.02 mm, respectively. For the higher wind speeds shear-induced turbulent heat exchange dominates, while for the lower wind speeds molecular and convective heat exchange dominate (Fairall et al., 1996a; Donlon et al., 1999; Donlon et al., 2002). The simulated warm layer effect was more intense for the seawater case than for the freshwater case, especially for low wind speeds (Figure 4 b, d). For wind speeds below 6 m s$^{-1}$, the difference was 0.03 ℃ in average, but could amount to almost 0.4 ℃. These differences do not take into account the effect of different solar absorption in the water column. Also, differences in the wave field and the atmospheric boundary layer in lakes could have an effect on skin temperatures (Wilson et al., 2013).



## 4.2 Cool skin effect

For high wind speeds, the different wind-dependent cool skin parameterizations converge to cool skin effect values of about -0.2 ℃ (Figure 5). The uncertainty is higher for low wind speeds, where convective and molecular heat transfer are more important than the effect of wind shear (Fairall et al., 1996a; Donlon et al., 2002) and the use of more complex

parameterizations is advised (Donlon et al., 2002). This is especially relevant for inland water bodies, where wind speeds are smaller than those found in the open sea (Wilson et al., 2013).

According to the results of the COARE algorithm, the mean estimated $\Delta T_c$ was -0.46 ℃, the maximum $\Delta T_c$ was -1.08 ℃ and the interquartile range was -0.31 ℃ to -0.61 ℃ (Figure 4 a). The estimated cool skin depth was mostly 2~3 mm (interquartile range: 1.8~2.8 mm), although occasionally the algorithm predicted skin depths deeper than 1 cm (maximum skin depth of

1.65 cm). During the day, the cool skin could disappear. These values are consistent with measurements at Lake Tahoe, with median skin effects of -0.34 ℃ to -0.46 ℃ at night (Wilson et al., 2013). Measurements taken in crater lakes (Oppenheimer, 1997) and cooling ponds (Wesely, 1979; Adams et al., 1990) show that cool skin effects can attain -1 ℃ to -3 ℃ in hot and heated inland water bodies.

## 4.3 Warm layer effect

The measured warm layer effect at 0.55 m was mostly positive during the day, with maximum values around midday (Figure 6). For wind speeds above 7 m s$^{-1}$, the daily variability in $\Delta T_w$(0.55 m) disappeared. This is consistent with a limit between 6 m s$^{-1}$ and 10 m s$^{-1}$ found by other authors (Gentemann et al., 2003; Gentemann and Minnett, 2008). The more important surface warming (of almost 2 ℃) occurred for low winds and high solar radiation, in accordance to many studies at the sea (Donlon et al., 1999; Donlon et al., 2002; Kawai and Wada, 2007; Gentemann and Minnett, 2008). The average observed

$\Delta T_w$(0.55 m) was 0.0 ℃, the maximum observed $\Delta T_w$(0.55 m) was 3.0 ℃, and the minimum $\Delta T_w$(0.55 m) was -0.7 ℃. At night $\Delta T_w$(0.55 m) was mostly negative, indicating the presence of a cool layer caused by convective cooling (Imberger, 1985).

Using the same data, as well as the data of the full length thermistor chain, we analysed the relation between the surface temperature (below the cool skin) and $T_{sml}$, the temperature of the surface mixed layer. The correlation between $T_{sml}$ and the

temperature at different depths was almost constant and near to 1 between the surface and a depth of 1.5 m (Figure 7). The RMSE we would incur on if we used any of these temperatures as an estimation of the SML temperature was less than 0.3 ℃. Below a depth of 2.5 m the RMSE sharply increased and the correlation coefficient sharply decreased, with an inflection point at a depth of about 5 m corresponding approximately to the seasonal thermocline depth.

In average the warm layer effect predicted by the COARE algorithm was small (mean of 0.25 ℃, interquartile range of

0.00~0.26 ℃), but it could attain maximum values of about 3 ℃ for wind speeds of 2~4.5 m s$^{-1}$ (Figure 4 b). These results contrast with the warm layer observations. The maximum values observed for this range of wind speeds were less than 2 ℃



in general (Figure 6) and the average observed warm layer effect was 0.0 ℃. Following Fairall et al. (1996a), we estimated the warm layer effect at the depth $z_r$ as

$$\Delta T_w(z_r) = \Delta T_w \frac{z_r}{D_T}, \tag{10}$$

where $D_T$ is the estimated depth of the warm layer, assuming a linear profile, and compared them to the field measurements. For $z_r = 0.55$ m, the coefficient of correlation between measurements and estimations was 0.45 and the root mean square error was 0.22 ℃, with the model overestimating the surface thermal gradient. The highest correlation between estimated and measured warm layer effect was found for $z_r = 2.50$ m, with a correlation coefficient of 0.78 and RMSE = 0.32 ℃. The discrepancies between the simulation and the measurements can be attributed to several factors:

   a) The model proposed by Fairall et al. (1996a) does not allow for negative warm layers (or cool layers). Part of the effects of convective cooling are included in the definition of the cool skin, but they are constrained to the upper millimetres of the water column.

   b) The meteorological data used to force the COARE algorithm was measured at about 10 km from the study site. Although air temperature data was corrected for altitude effects, local micrometeorological conditions at the reservoir of Bimont are probably different to those measured at the meteorological station of Aix-en-Provence (Prats et al., in press). For example, the wind field might be affected by local orography and relative humidity could be affected by the presence of the reservoir.

   c) The parameterization of the solar radiation absorption might be inadequate. Ohlmann and Siegel (2000) proposed a parameterization of the transmission of solar radiation in the ocean which took into account the effect of the vertical distribution of chlorophyll in the water column, the cloud amount and the solar zenith angle. The implementation of this parameterization into the COARE bulk flux algorithm, improved the quality of the sea surface temperature simulations (Ohlmann and Siegel, 2000; Wick et al., 2005). The COARE 3.0 solar absorption parameterization might be particularly ill-adapted to the freshwater case, since freshwater is typically less transparent than open sea water.

   d) The distortion of the temperature field by the measuring device and platforms may cause errors in surface temperature measurements (Kawai and Kawamura, 2000). Kawai et al. (2009) found differences exceeding 1.0 ℃ between measurements taken at a depth of 0.20 m at different sides of a surface-moored buoy 2.4 m in diameter. The differences were greater when diurnal warming was important. Since the buoy used for the 0.55 m chain (a polystyrene plate 0.75 m wide) was much less intrusive, we expect the platform effect to be less important. In addition, a comparison between the 0.55 m sensor of the 0.55 m chain and the 0.5 sensor of the nearby full length chain revealed only small differences.



## 5. Quality assessment of LakeSST

### 5.1 Effect of the time of measurement

In the previous sections we analysed the relation between the surface temperature and $T_{0.55m}$ all day round. However, the time of Landsat pass above a given Earth zone is quite constant. During the study period, the Landsat satellites passed above

the reservoir of Bimont between 9:54 UTC and 10:26 UTC. In this section we analyse $\Delta T_w(0.55m)$ at 10:00 UTC.

At 10:00 UTC, $\Delta T_w(0.55m)$ was very small for most measurements at the reservoir of Bimont (Figure 8): the mean $\Delta T_w(0.55m)$ was 0.05 ℃, the median $\Delta T_w(0.55m)$ was 0.00 ℃ and the standard error was 0.21 ℃. The maximum observed value of $\Delta T_w(0.55m)$ at 10:00 UTC was 1.6 ℃, but $\Delta T_w(0.55m)$ was higher than 0.5 ℃ only 5 % of the time. Considering that an uncertainty of at least 0.3 ℃ is expected in optimal conditions for satellite-derived surface temperature estimations

(Jiménez-Muñoz and Sobrino, 2006), satellite temperatures measured at about 10:00 UTC may be accurate enough for many ecological applications. However, if heat or gas fluxes are to be estimated from satellite measurements, they should be corrected to account for the cool skin and warm layer effects.

### 5.2 Precision of satellite-based surface water temperature measurements

The Figure 9 shows the differences in temperature between successive scenes for individual pixels. The differences in

temperature for individual pixels in successive scenes is unbiased with a median difference of 0.00 ℃ (mean: -0.08 ℃; interquartile range: -0.0007 ℃ to 0.0007 ℃) and a root mean square error of 0.20 ℃. There is a feeble variation of the RMSE between groups, whether the data is grouped by month (0.28 ℃ to 0.49 °C, Figure 9d), season (0.36 ℃ to 0.40 °C, Figure 9e) or year (0.25 ℃ to 0.52 °C, Figure 9f). In contrast, occasionally there can be important variations of temperature differences within a given group (*e.g.* -2.4 ℃ to +7.9 ℃ for the month of July, Figure 9a). Temperature differences remain

within the range from -2 ℃ to +2 ℃ for most years. However, in some notable cases (especially the years 2007 and 2011, Figure 9c) the maximum differences could exceed 5 ℃. The most important biases (more than 3 ℃) were concentrated on a few particular dates and water bodies (Table 2). In all cases, these biases were localised near the water edge (Figure 10), often on the West side of the water body, but sometimes also on the East side. The problem is due to georeferencing errors. The Lac de Carcans-Hourtin shown in the Figure 10 is located at low altitude and near the sea and the image with path/row

201/029 shows a horizontal displacement of several hundred meters. Geometric accuracy of Landsat images varies with instrument and processing level. For example, Landsat ETM+ nominal accuracy is at least 250 m 90 % of the time at the sea level in areas of low relief, while Landsat 5 TM level L1GS images have a geometric accuracy of at least 700 m 90 % of the time at the sea level in areas of low relief. Georeferencing errors may result in important errors in the temperatures retrieved from satellite images (Sentlinger et al., 2008).



### 5.3 Satellite images artefacts

The Landsat website (https://landsat.usgs.gov/known-issues) lists several types of artefacts that can be present in the Landsat data. The artefacts that can affect thermal readings include banding, impulsive noise, coherent noise and memory. Banding consists in the apparition of bands in the images that can be due to different types of error. Impulsive noise can include both underestimation and overestimations of radiance in individual pixels. It is often linked to problems of transmission, treatment of transcription of the images, but may be due to several reasons. Coherent noise appears in the form of a repetitive noise pattern in the images, which can be caused by different electric systems on board of the satellite. Memory effects are due to the reduced response of the sensor after scanning a bright (hot) target. As a result, if after the bright (hot) target the sensor finds an uniform region like the water surface, the measured values will be slightly lower than in the next scan in the inverse sense, producing a banding (Teillet et al., 2004; NASA, 2011). This artefact could affect surface temperature measurements in water bodies, especially when there are great differences with the surrounding terrain. The thermal band measurements of Landsat 7 ETM+ can be slightly affected by memory effects (Goward et al., 2001). The banding in Landsat 7 images is corrected during image production wherever it is detectable, *i.e.* in smooth, homogeneous surfaces (ESA, 2003; NASA, 2011). According to Teillet et al. (2004), memory effects have not been observed in the measurements of the thermal band of Landsat 5 TM.

We have no measure of the incidence of these artefacts in the data base, but we have observed banding in some images (e.g. Figure 10). Also, in the Figure 10 temperature varies spatially in an East-West direction. Such spatial variability may be due to upwellings caused by the wind (Marti-Cardona et al., 2008). However, the presence of banding may indicate that memory effects are also present. To decrease the impact of artefacts, we recommend using the median temperature for each satellite image. The advantage of using the median is that it is a robust statistic resistant to the effect of extreme values. To validate the spatial variability of temperature in satellite images, several spatially distributed measurements points in a water body would be necessary.

### 5.4 Comparison of satellite measurements to field data

The correlation between satellite temperatures and in situ SML temperatures was good ($\rho$=0.96~0.97) and the overall RMSE was 1.5 ℃ (Table 1, Figure 11). Satellite measurement underestimated $T_{0.50m}$ by 0.79 ℃ and $T_{sml}$ by 0.49 ℃. The difference in bias between $b_{0.50m}$ and $b_{sml}$ was statistically significant (p-value < 2.2e-16). For the SML temperature, the underestimation is less important, since the SML temperature is slightly lower than the surface temperature in a stratified water body. Although the bias for AUL13 and LPC38 seems more negative than the rest, a one-way ANOVA did not reflect statistically significant differences between water bodies. These observations confirmed the uncertainty estimations by the work preceding this study (Simon et al., 2014), with coefficients of determination above 0.9 and RMSE of 1~2 ℃ and a tendency of satellite images to underestimate bulk surface temperatures.



The error statistics found for French water bodies are comparable to those found in other studies using satellite images to estimate surface water temperatures. Crosman and Horel (2009) found a cool bias of -1.5 °C of MODIS-derived surface temperatures of the Salt Lake relative to $T_{0.5m}$. Lamaro et al. (2013) used Landsat 7 ETM+ thermal images to obtain surface temperatures for the Río Tercero reservoir in Argentina. They obtained an RMSE of 1.2 °C when using the SC algorithm of (Jiménez-Muñoz et al., 2009) and an RMSE of 1.0 °C when using the Radiative Transfer Method.

The bias showed a seasonal pattern, with underestimations occurring in the months of September to January (Figure 12). Lamaro et al. (2013) observed that the SC algorithm tended to overestimate water temperatures except in July (winter in the Southern hemisphere) when they were underestimated. They think it might be caused by the greater difference between the colder air and the water. Allan et al. (2016) showed that depending on the sources of atmospheric data the SC algorithm tended to underestimate or overestimate water surface temperatures.

To find the reason for the observed seasonal bias, we calculated the correlation between $b_{0.50m}$ and several variables: water temperature, atmospheric water content, distance between the measuring station and the nearest valid pixel, daily solid precipitation, daily liquid precipitation, daily air temperature, daily specific humidity, daily wind speed, daily downwelling longwave radiation, daily incoming solar radiation, solar elevation at 10 a.m. and daily clearness index. To avoid multiple testing issues when assessing the statistical significance of correlations, we used a Bonferroni correction of the significance level using $\alpha = 0.05/13$ where 13 is the number of tests. The bias was correlated with air temperature ($\rho = 0.38$, p-value = 1.7e-4), daily solar radiation ($\rho = 0.54$, p-value = 1.2e-8), solar elevation at 10 a.m. ($\rho = 0.44$, p-value = 5.7e-7) and clearness index ($\rho = 0.47$, p-value = 1.5e-6). Similar results could be found for $b_{sml}$. Since some of these variables were correlated among themselves, the number of variables to explain the variability in the bias could be reduced to just two by using stepwise model selection by Akaike information criterion:

$$b_{0.50m} = -12.66 + 6.608 \cdot 10^{-4} R_s + 31.07 k_t - 22.09 k_t^2 \tag{11}$$

$$b_{sml} = -13.08 + 8.380 \cdot 10^{-4} R_s + 33.04 k_t - 24.04 k_t^2 \tag{12}$$

where $k_t$ is the clearness index and $R_s$ is the downwelling solar radiation (J cm$^{-2}$). The Figure 13 shows the dependence of bias on $R_s$ and $k_t$. The bias was more important for $k_t < 0.6$. For $k_t \geq 0.6$, the average bias was $b_{0.50m} = $ -0.47 °C and $b_{sml} = $ -0.11 °C, consistent with a cool skin effect and a negligible average value of $\Delta T_w(0.55m)$ (Section 4).

The effect of $k_t$ on the temperature estimations may be explained as the effect of clouds. The clearness index is well correlated with atmospheric emissivity: for overcast skies and clearness index near zero the emissivity reaches a maximum of about 0.95; for clearer skies, emissivity decreases (Aubinet, 1994). In addition, although not statistically significant at the Bonferroni-corrected $\alpha$ level, there were feeble correlations of bias with liquid precipitation ($\rho = $ -0.21, p-value = 0.038) and the percentage of valid pixels ($\rho = 0.28$, p-value = 0.0024), which seems to confirm the effect of clouds. In fact, the SC algorithm was calibrated for clear-sky conditions (Jiménez-Muñoz and Sobrino, 2003), but we applied the algorithm also when clouds were partially present. Clouds increase the downwelling longwave radiation, decreasing the longwave radiation at the top of the atmosphere (Hatzianastassiou and Vardavas, 1999). This explains the more important underestimations for lower $k_t$. Additionally, the effect of clouds on downward longwave irradiance is often parameterized as a quadratic function





of cloud cover (Iziomon et al., 2003). The atmospheric emissivity $\varepsilon_a$ can be parameterized as a function of clear-sky atmospheric emissivity $\varepsilon_{clr}$ and cloud cover $c$ as

$$\varepsilon_a = (1 + ac^b) \tag{13}$$

where $a$ and $b$ are constants, and $b$ can take values between 1 and 2.75 (Flerchinger et al., 2009).

Different reasons may explain the inclusion of $R_s$ in Eq. (11) and Eq. (12):

    a)   The term $R_s$ may be related to cloud effects not accounted for completely by $k_t$.

    b)   Given the high correlation between $T_a$ and $R_s$ ($\rho = 0.69$, p-value = 2e-16), the term $R_s$ may partially account for the air temperature effects. Although not correlated to emittance, air temperature is well correlated to sky temperature (Aubinet, 1994), affecting the longwave radiation balance. The dependence between air temperature and longwave
radiation balance is not taken into account by the SC algorithm. In fact, the results of SC algorithms, including the one used here, can be improved by using $T_a$ in addition to water vapour as input data (Qin et al., 2001; Cristóbal et al., 2009).

## 6. Applications

Satellite data can be used to demonstrate spatial and temporal patterns of surface water temperature. A first analysis of
spatial patterns was carried by Prats and Danis (2015) using a preliminary version of the database and showing a good correlation between surface temperature and latitude and altitude. Since satellite water temperature measurements are a continuous source of data, they can be used for long term studies, such as the monitoring of climate change effects. This was demonstrated by Schneider and Hook (2010) and Schneider et al. (2009) by calculating surface temperature trends obtained from AVHRR (Advanced Very High Resolution Radiometers) and ATSR (Along Track Scanning Radiometer) instruments.
Such studies have demonstrated warming rates of 0.11 °C yr$^{-1}$ for lakes in California and Nevada in over the period 1992-2008 (Schneider et al., 2009) and an average of 0.045 °C yr$^{-1}$ in over the period 1985-2009 for 167 large inland water bodies worldwide (Schneider and Hook, 2010).

Satellite measurements can be used to complement the data used for the calibration and validation of hydrodynamic and water quality models of lakes. Although these models require profile data, long data series of profile data are still rare and
satellite measurements can be useful to provide long term data for the study at hand. The value of the calibration parameters depends on the available data (Andréassian et al., 2012; Prats and Danis, 2017) and it is interesting to test the long term performance of a model, especially if it will be used to predict the effects of climate change or similar long term effects. Satellite measurements have been used to assess the long-term performance of a hydrodynamic model of the reservoir of Bimont (Prats et al., in press). Satellite images are particularly useful for the application of 2D and 3D models.
Satellite images can be used to estimate surface fluxes. The knowledge of the surface fluxes of heat and gases. is important for research and management. The study of the thermal surface fluxes is important to understand the thermal behaviour of lakes (Henderson-Sellers, 1986). A good estimation of evaporation is necessary to accurately estimate the hydrologic budget



of a lake (Sahoo et al., 2013) and allows an estimation of a part of the water footprint of a nation due to artificial water reservoirs (Hoekstra, 2017). These fluxes depend on $T_{int}$ (Saunders, 1967; Kawai and Wada, 2007); $T_{int}$ determines the emitted longwave radiation and the value of the saturation vapour pressure at the surface, affecting latent and sensible flux calculations (Webster et al., 1996). The interface temperature $T_{int}$ is a theoretical value and cannot be known (Donlon et al., 2002). In the absence of such a value, experimental studies often use some form of surface bulk temperature to estimate the heat fluxes of inland water bodies (e.g. Oswald and Rouse, 2004; Binyamin et al., 2006; Ramos-Fuertes et al., 2016). However, sea models show that the skin effect on surface heat flux calculations can be of the order of a few tens of watts per square meter (Fairall et al., 1996a; Webster et al., 1996) and it can also affect the exchange of gases (Kawai and Wada, 2007). Satellite images can be used to improve the accuracy of the estimations (Lofgren and Zhu, 2000), assuming that $T_{skin}$ is close enough to $T_{int}$ (Kawai and Wada, 2007).

## 7. Data availability

The LakeSST data set is distributed under a Creative Commons Attribution 4.0 License. The data may be downloaded from the data repository Zenodo at https://doi.org/10.5281/zenodo.1041746. The original temperature ASCII rasters used to derive the data set may be obtained by request from the authors.

## 8. Conclusion

The LakeSST data set contains skin surface water temperature data for 442 French water bodies for the period 1999-2016 obtained from archives of Landsat 5 and Landsat 7 thermal infrared images. The overall accuracy of the satellite-derived temperature measurements is about 1.5 ℃, similar to other applications of satellite images to estimate freshwater surface temperatures. The spatial and temporal coverage of the database make it an ideal resource for studies on the long term evolution of lake surface temperature and for geographical studies of temperature patterns.

Since cool skin and warm layer algorithms have been developed mainly for the sea and have not been extensively tested in a freshwater environment —except in extreme cases such as cooling ponds (Wesely, 1979; Adams et al., 1990)—, we provide the satellite temperature data "as is", without applying any correction, leaving the user free to decide which correction to apply, if necessary. In fact, in some cases (e.g. studies on surface fluxes) a correction may not be necessary. If a cool skin correction is used (*i.e.*, if the user is interested in the temperature of the first 1~2 m), we recommend using one of those which can take into account the differences in the physical characteristics between freshwater and seawater and an appropriate parameterization of the solar radiation absorption in the water column.

We suggest using the median temperature for each image as an estimation of the average surface temperature of each water body on the measurement date. The SC algorithm tends to underestimate surface temperature on cloudy conditions. This





problem may be solved by applying Eq. (11) and Eq. (12) or by discarding all the measurements on which the daily clearness index was less than 0.6.

*Author contribution*. T. Tormos supervised the downloading and processing of satellite images and designed the methodology. N. Reynaud contributed to download satellite images, maintain and improve script processing and performance, and process images. D. Rebière and T. Peroux designed, installed and operated the 0.55 m thermistor chain at the reservoir of Bimont and the full length thermistor chains of the continuous lake monitoring network. J. Prats calculated summary statistics of the satellite images and prepared the dataset LakeSST. J. Prats and P.-A. Danis made the quality assessment of satellite images. J. Prats prepared the manuscript with contributions from the other co-authors. All authors have read and approved the final manuscript.

*Competing interests*. The authors declare that they have no conflict of interest.

*Acknowledgements*. This study was funded by the Onema (French National Office for Water and Aquatic Environments) action 61B, 102 and CK. The authors thank R. Simon for his contribution in preparing the script to process satellite images. The authors thank Météo-France for the meteorological data at the station of Aix-en-Provence and the data from the SAFRAN analysis, provided free of charge to Irstea for this study and for the project "Temperature and oxygen indicators for lakes". The authors are grateful to ECMWF and USGS for providing the data used in producing the surface temperature data set. Special thanks to Julien Dublon, Thierry Point, Jean-Michel Foissy, Michael Cagnant, Jean-Claude Raymond, William Sremski, Gaël Olivier, Sylvain Richard, Laurent Tachot and the agents of the Departmental Services of the French Agency for Biodiversity for the Departments 04, 13, 38 and 63.

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



**Table 1: Comparison of satellite-based temperature estimations to *in situ* measurements.**

| Water body code | Water body name | Period | Number of meas. | $T_{0.01\,m} \sim T_{sml}$ | | | $T_{0.01\,m} \sim T_{0.50m}$ | | |
|---|---|---|---|---|---|---|---|---|---|
| | | | | ME [ºC] | R² | RMSE [°C] | ME [ºC] | R² | RMSE [°C] |
| AUL13 | Aulnes | 24/09/2013-10/09/2014 | 5 | -1.08 | 0.99 | 1.58 | -1.22 | 0.99 | 1.56 |
| BIM13 | Bimont | 24/02/2014-30/12/2016 | 34 | -0.32 | 0.95 | 1.47 | -0.56 | 0.96 | 1.45 |
| LPC38 | Pierre-Châtel | 23/08/2013-27/05/2016 | 13 | -1.32 | 0.98 | 1.66 | -1.54 | 0.98 | 1.86 |
| PAV63 | Pavin | 24/10/2013-28/09/2015 | 16 | -0.52 | 0.98 | 1.17 | -0.69 | 0.97 | 1.25 |
| SCR04 | Sainte-Croix | 31/07/2013-30/12/2016 | 49 | -0.32 | 0.96 | 1.54 | -0.74 | 0.97 | 1.52 |
| All lakes | | 23/08/2013-30/12/2016 | 117 | -0.49 | 0.96 | 1.49 | -0.79 | 0.97 | 1.51 |





**Table 2: List of pairs of overlapping satellite images with temperature differences higher than 3 ºC.**

| Date | Satellite | Path/Row | Water body |
|------|-----------|----------|------------|
| 12/05/2002 | Landsat 7 | 197/030, 197/031 | Lac de Matemale (MAT66) |
| 22/09/2004 | Landsat 7 | 197/030, 197/031 | Lac de Puyvalador (PUY66) |
| 12/04/2007 | Landsat 5 | 201/028, 201/029 | Lac de Carcans-Hourtin (ECH33) |
| 22/03/2011 | Landsat 5 | 201/028, 201/029 | Lac de Carcans-Hourtin (ECH33) |
| 07/04/2011 | Landsat 5 | 201/028, 201/029 | Lac de Carcans-Hourtin (ECH33) |
| 29/08/2011 | Landsat 5 | 201/028, 201/029 | Lac de Carcans-Hourtin (ECH33) |



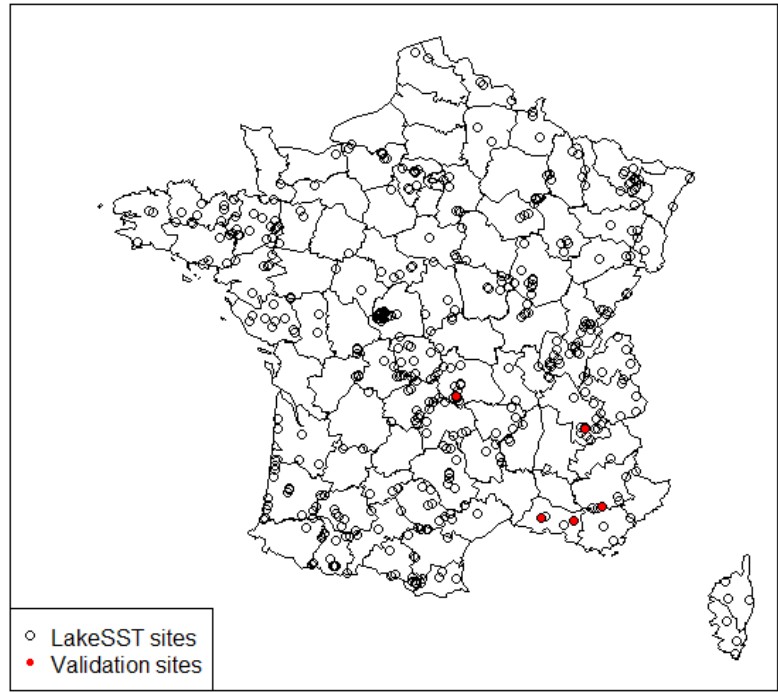

Figure 1: Location of the water bodies included in LakeSST and validation sites.




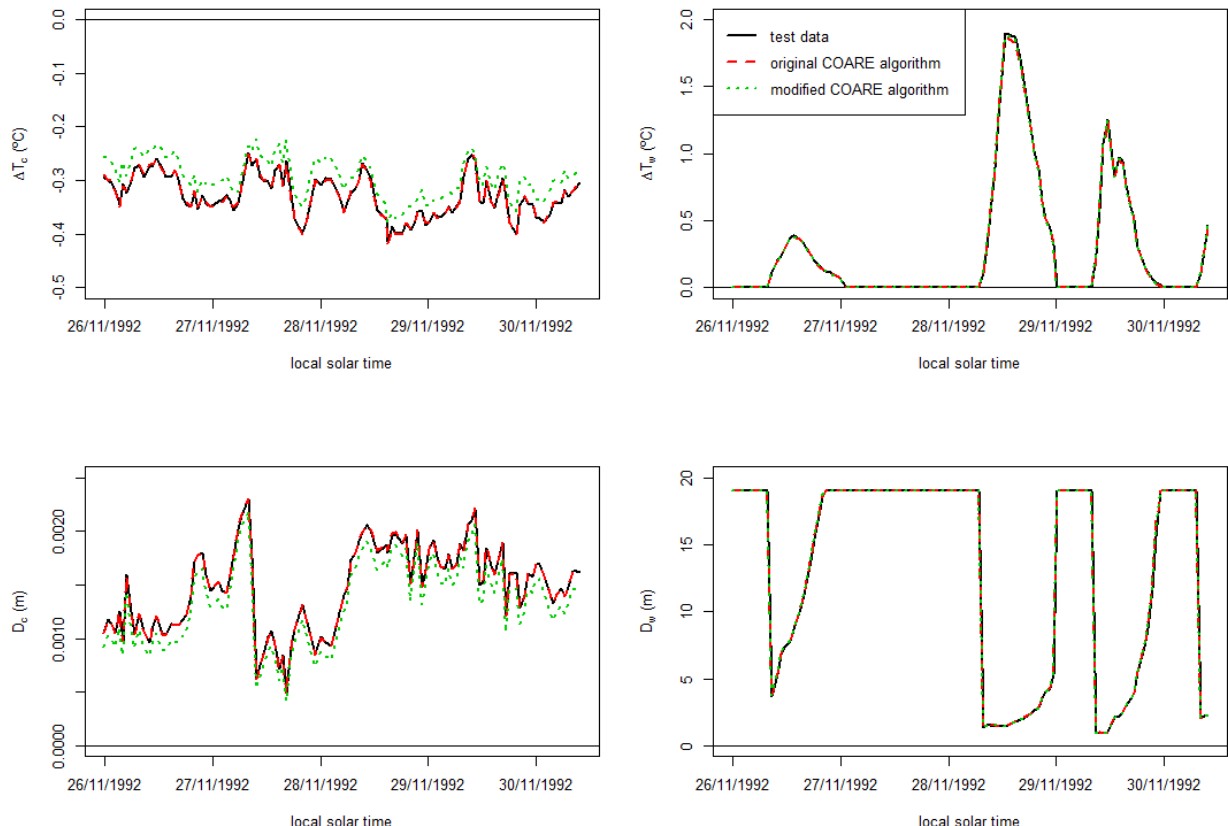

**Figure 2: Validation of the implementation of bulk flux COARE algorithm and its modification: cool skin effect ($\Delta T_c$), cool skin depth ($D_c$), warm layer effect ($\Delta T_w$) and warm layer depth ($D_w$).**





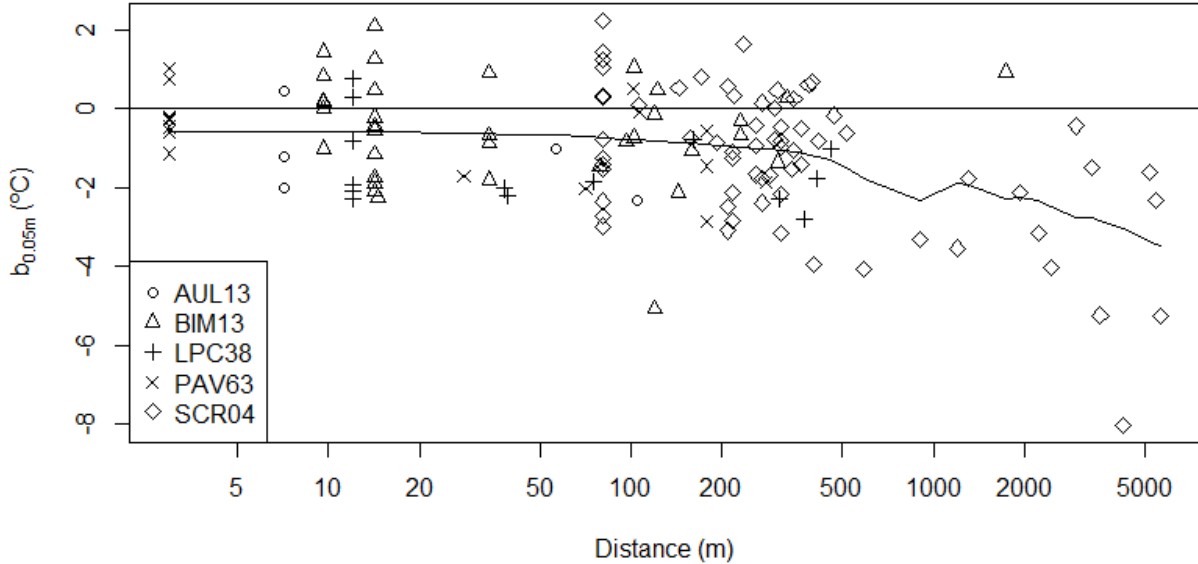

Figure 3: Measurements and LOWESS of the bias of $T_{0.50\ m}$ as a function of the distance between the measurement point and the nearest valid pixel in the satellite image.





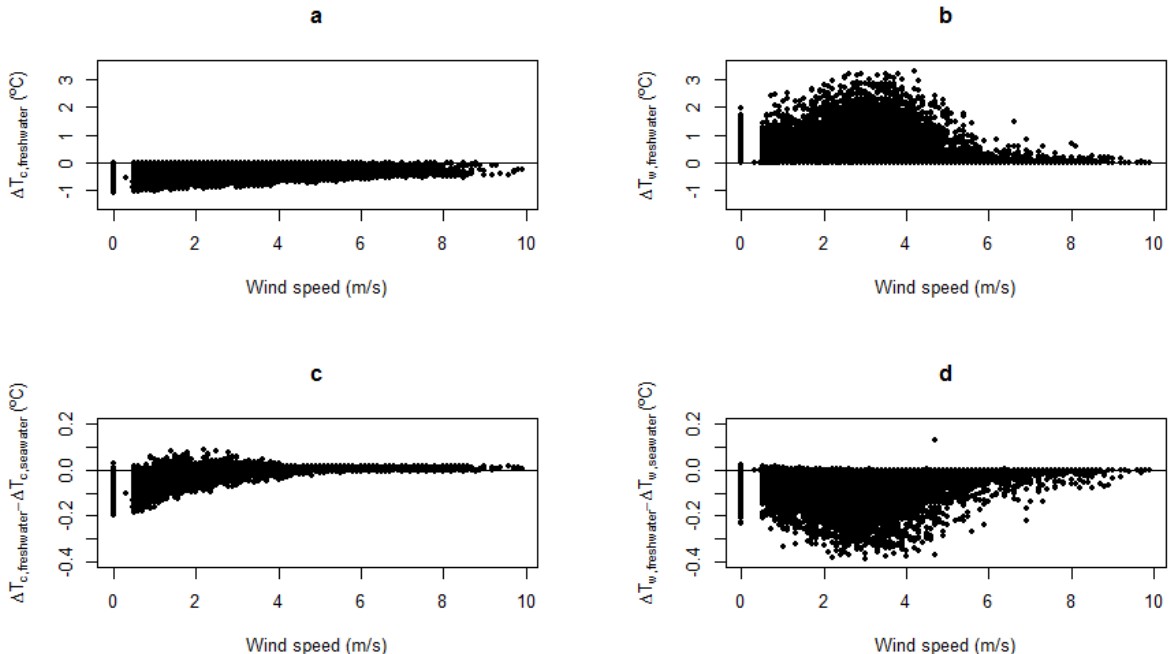

**Figure 4: Cool skin (a) and warm layer (b) effect estimated by the model proposed by Fairall et al. (1996a) using the data for Bimont; and difference of cool skin (c) and warm skin (d) effect between freshwater (salinity=0 g/kg) and seawater (35 g/kg) according to the same model and meteorological conditions.**

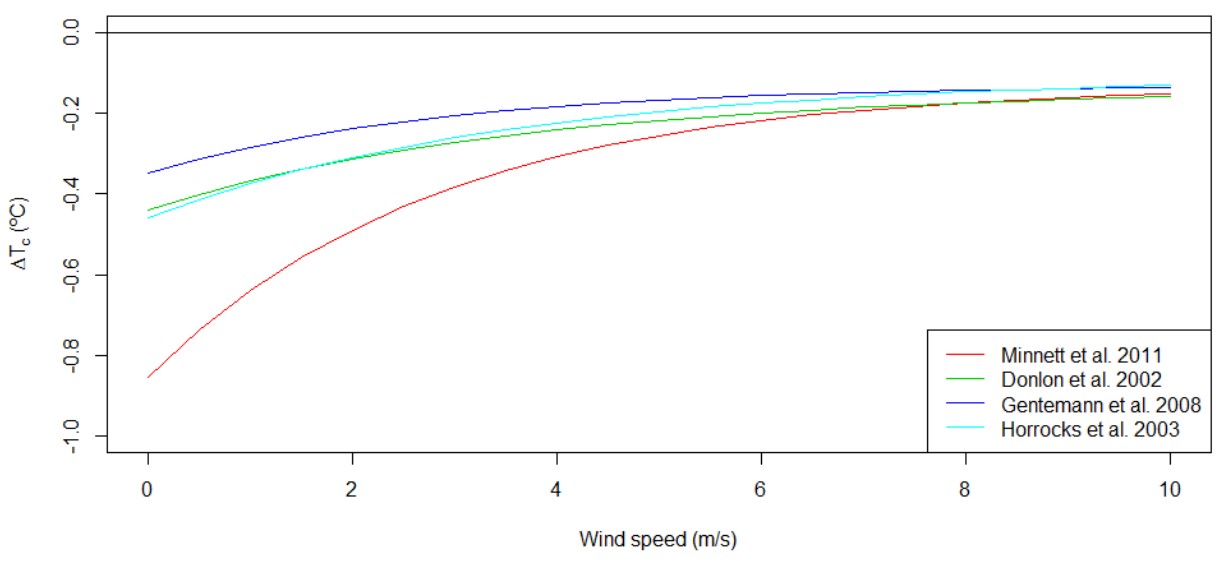

**Figure 5: Cool skin effect ($\Delta T_c$) as a function of wind speed according to several empirical parameterizations.**




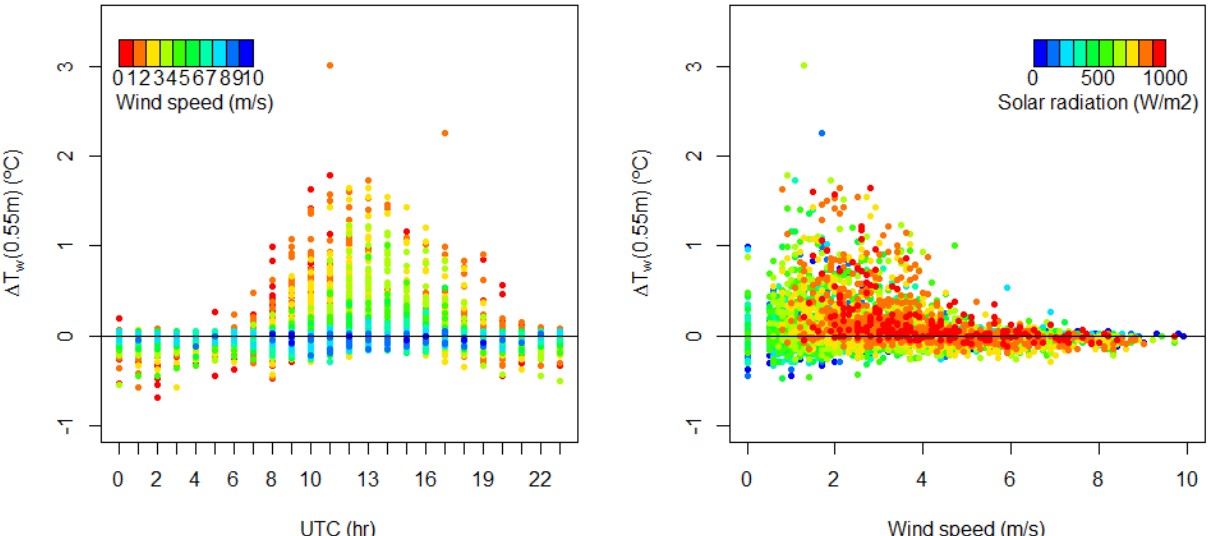

**Figure 6: Warm layer effect ($\Delta T_w$) at 0.55 m at the reservoir of Bimont as a function of the time of measurement, wind speed and solar radiation, 21 February 2014 to 31 July 2016.**





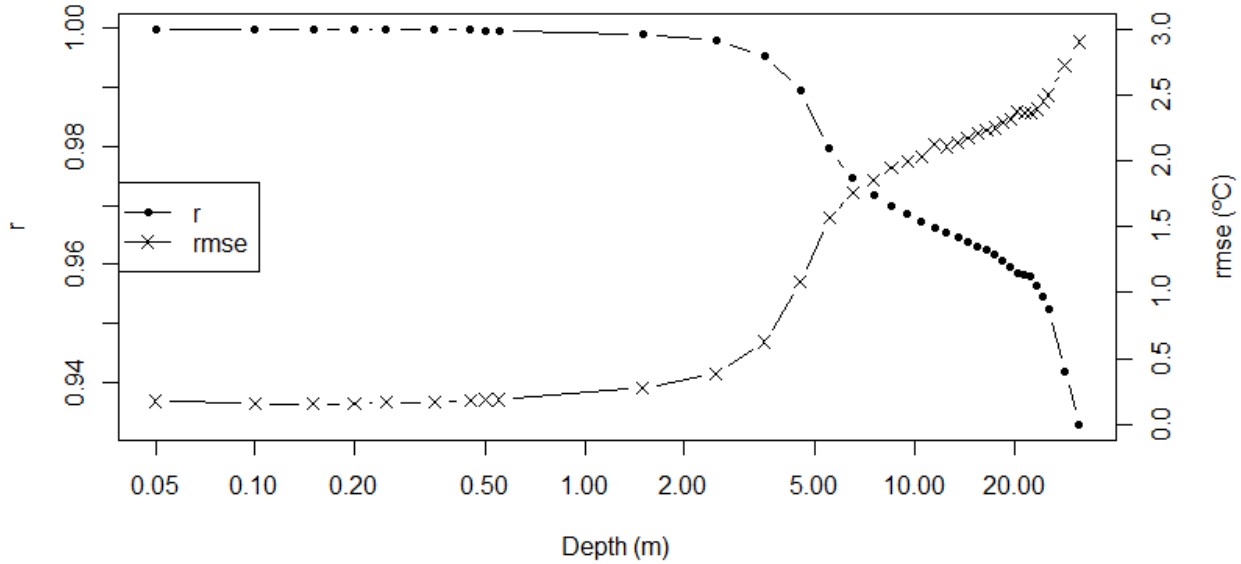

**Figure 7: Coefficient of determination and RMSE of the SML temperature in relation to temperature at other depths.**

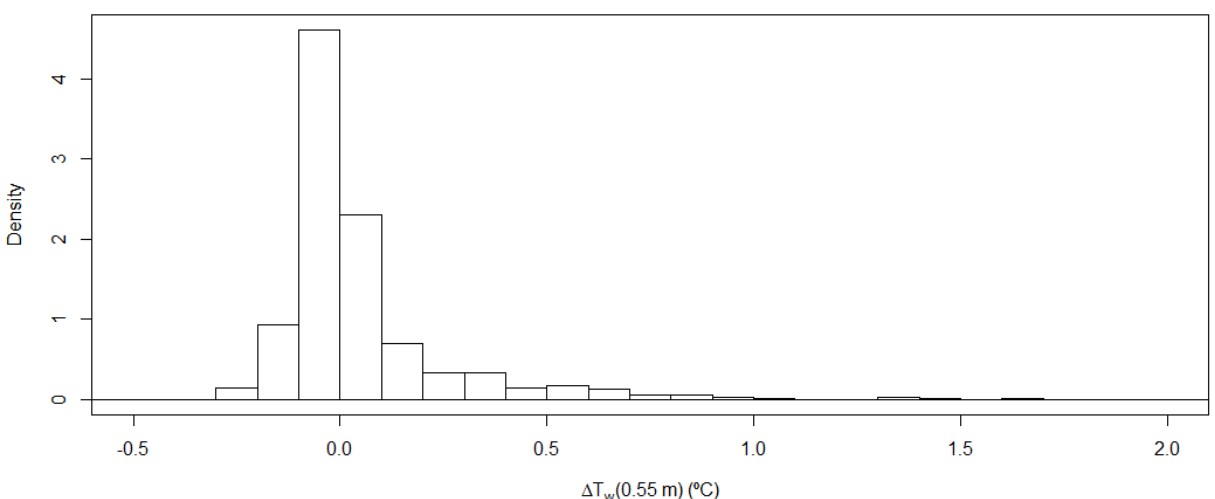

**Figure 8: Histogram of $\Delta T_w$(0.55m) at the reservoir of Bimont at 10:00 UTC, 21 February 2014 to 24 April 2016.**



**Figure 9: Above:** boxplots of the differences in temperature between consecutive scenes for individual pixels grouped by month of the year (a), by season (b) and by year (c). **Below:** Root mean square error of the temperature differences grouped by month of the year (a), by season (b) and by year (c).





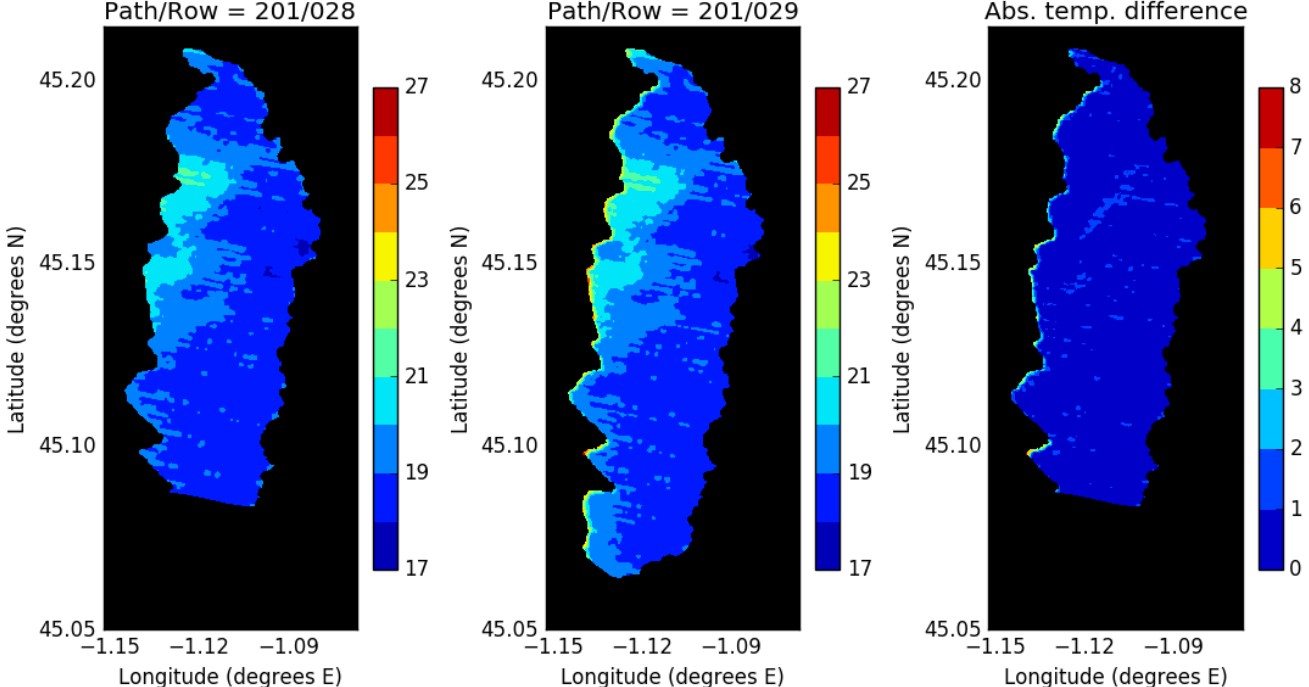

**Figure 10: Water temperature according to two overlapping images for the Étang de Carcans-Hourtin (ECH33) on April 7th 2011, and absolute temperature difference between both images.**





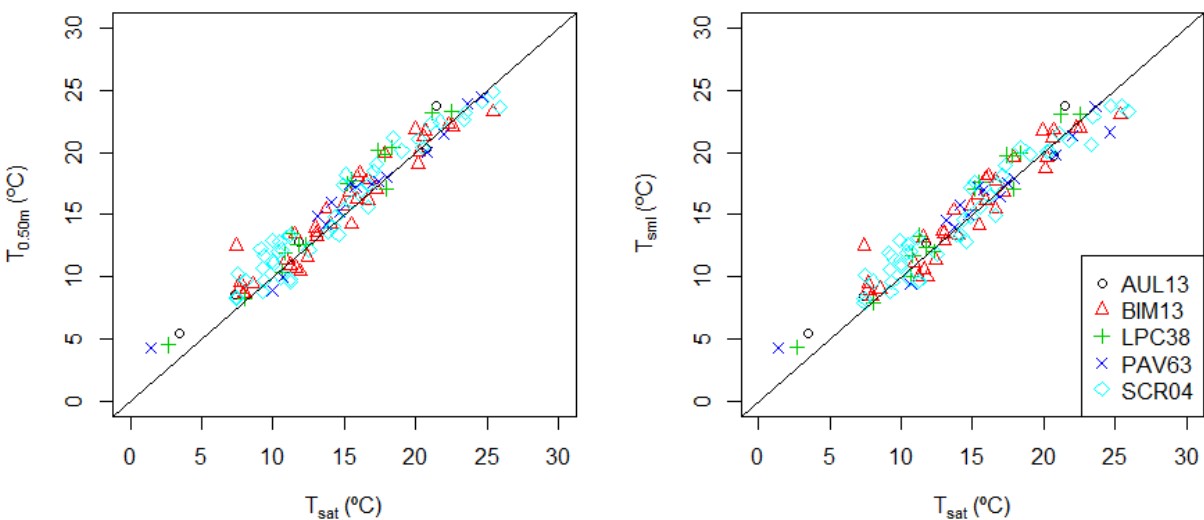

**Figure 11: Comparison of satellite-based temperature estimations and *in situ* temperatures: a) temperature at 0.5 m, b) temperature of the surface mixed layer.**

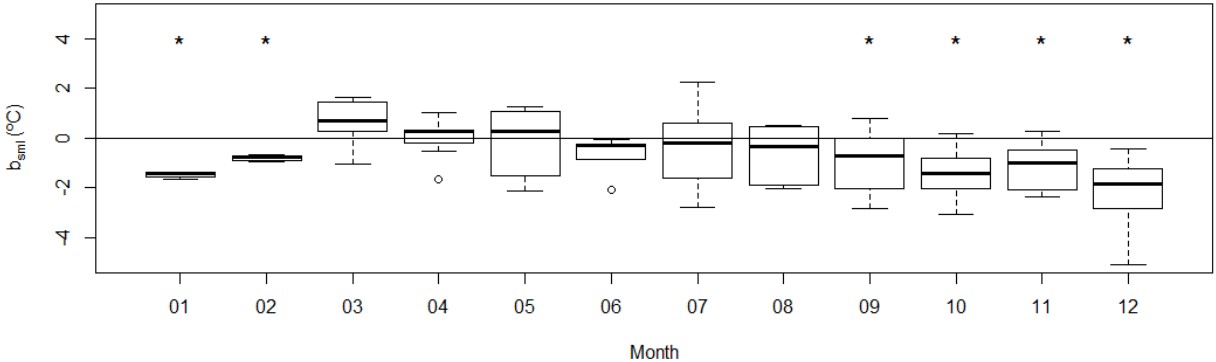

**Figure 12: Bias for the temperature at 0.50 m as a function of the month of the year. Asterisks indicate statistically significant differences from zero at the 0.05 level.**



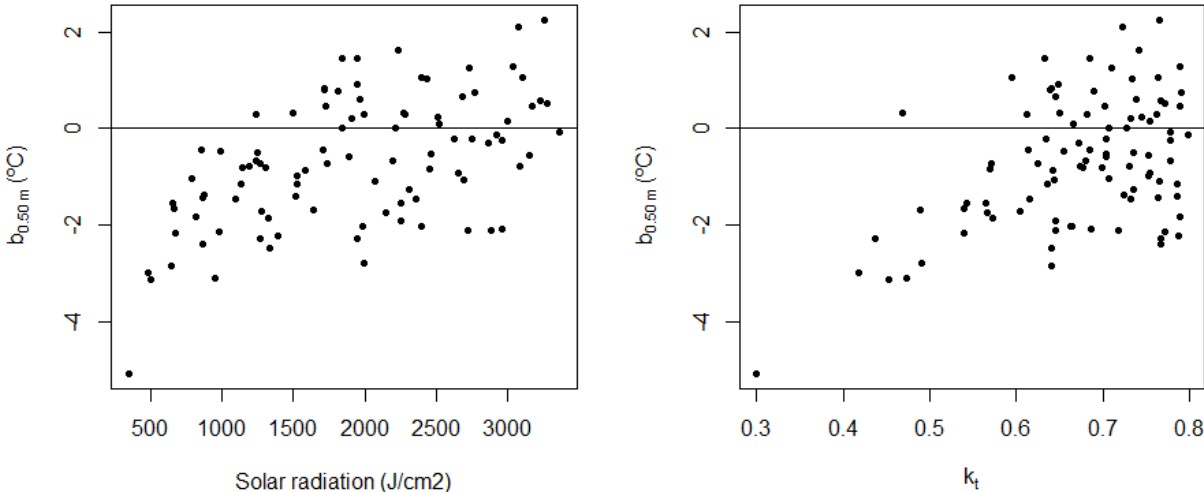

**Figure 13: Dependence of the bias of satellite images in relation to $T_{0.50m}$ ($b_{0.50m}$) on solar radiation and the clearness index ($k_t$).**