# Peer review of "LakeSST: Lake Skin Surface Temperatures in French inland water bodies for 1999-2016 from Landsat archives"

_Earth System Science Data, 2017_

## Referee Comment (RC1) · Anonymous Referee #1 · 22 Dec 2017

This paper presents a useful dataset of remotely sensed surface temperature (Landsat series) over French water bodies, including also some ancillary data. The dataset is described in detail, and one of the most remarkable things is the detailed discussion on the problems related to the direct comparison between temperatures retrieved from satellites (skin) and water temperatures measured by contact at different depths. The paper also shows a rigorous validation of the dataset. Therefore, this paper address the main topics of this journal, and my recommendation is to publish the paper after some minor comments:

GENERAL COMMENTS

[Figure]

As a general comment, I think the validation part could be improved (if not now, then in a future work). I would be nice to measure with a thermal radiometer over particular lakes and then compared contact temperatures with radiometric temperatures. Radiometric temperatures are also better indicator of the performance of the surface temperature retrieval from satellite data. Another option is to perform an intercomparison between standard and well-validated remote sensing SST/LST products. In this case most of the products available are at low spatial resolution (around 1-km), so this intercomparison may be restricted to the largest lakes.

It should be also justified why Landsat-8 is not used, since the straylight problem in the TIR bands was partially solved.

SPECIFIC COMMENTS:

-Section 3.2, page 6, line 11: If I am not wrong the algorithm used by Simon et al 2014 is the same than the algorithm presented by Jimenez-Munoz & Sobrino 2009, so it is not a new version.

-Section 5.3, page 13: Please include some comments about the SCL-off problem in Landast-7, and how this problem is addressed in the presented dataset. If Landsat-8 is not used for any technical resason (e.g. straylight problem), then some sentence could be also added in this section.

-Section 5.4, page 14, lines 31-32: "... we applied the algorithm also when clouds were partially present." This is a critical issue, because all the surface temperature retrieval algorithms working with Thermal-Infrared data are developed to be applied under clear sky conditions. This is well-known, so I think it has no sense to apply the algorithm in the presence of clouds. I would remove the data points contaminated by clouds and redo again the analysis to assess which are the main variables contributing to the seasonal bias. May be it is related to different atmospheric water vapor contents (?)

-Table 1: The header of the table should be more informative.
-References: in page 6 line 7-8 the authors refer to Sobrino 2004 with a strange symbol (#2738), but Sobrino 2004 is not included in the references list. Please correct.

---

## Referee Comment (RC2) · Anonymous Referee #2 · 3 Jan 2018

My recommendation is to publish the paper after some minor comments have been addressed:

Accuracy of 1.5 degrees is problematic. This is a very large discrepancy. Can you justify why this is sufficient for climate change studies?

P2. L6: "4" should be "four". Any number less than 10 should be spelled out.

Measurements with a thermal radiometer over the water bodies would be ideal. Then a more robust comparison of temperatures with radiometric temperatures is possible. Radiometric temperatures are a much better indicator of the performance of the surface temperature retrieval from satellite data.

Include some comments about the SCL-off problem in Landast-7, and how this problem is addressed in the presented dataset. Discussion about why was Landsat 8 not included is necessary.

I would highly recommend consultation with a native English speaker to clarify the language used. For example: P2. L13-15: this statement makes no sense.

Much effort is given to the evaluation of various algorithms but more effort is needed to justify the use of daytime satellite retrievals and the impacts of the skin effect.

Why include the numerous outlines of various french regions in Figure 1? These are not defined in the legend and only distract from the overall presentation of the figure.

Fonts on the y-axis of Figure 4 are quite small and difficult to read.

Figure 11 should include R2 and p values

---

## Author Comment (AC1) · 8 Feb 2018

*Please find our response in italics.*

**RESPONSE TO ANONYMOUS REFEREE #1**

This paper presents a useful dataset of remotely sensed surface temperature (Landsat series) over French water bodies, including also some ancillary data. The dataset is described in detail, and one of the most remarkable things is the detailed discussion on the problems related to the direct comparison between temperatures retrieved from satellites (skin) and water temperatures measured by contact at different depths. The paper also shows a rigorous validation of the dataset. Therefore, this paper address the main topics of this journal, and my recommendation is to publish the paper after some minor comments:

GENERAL COMMENTS

As a general comment, I think the validation part could be improved (if not now, then in a future work). I would be nice to measure with a thermal radiometer over particular lakes and then compared contact temperatures with radiometric temperatures. Radiometric temperatures are also better indicator of the performance of the surface temperature retrieval from satellite data. Another option is to perform an intercomparison between standard and well-validated remote sensing SST/LST products. In this case most of the products available are at low spatial resolution (around 1-km), so this intercomparison may be restricted to the largest lakes.

*We totally agree with reviewer #1 but we do not have surface temperature data measured by radiometry yet. We are reflecting on how to obtain such data either by acquiring radiometric measurements through collaborations or by obtaining funding for the necessary instrumentation and field work.*

It should be also justified why Landsat-8 is not used, since the straylight problem in the TIR bands was partially solved.

*When this study was started, the issues with Landsat 8 TIR bands made us discard Landsat 8 data. Besides, the algorithm correcting stray light effects was only implemented into the Landsat processing*

*system on February 2017. At the moment, some of the authors, in collaboration with other researchers, are testing different algorithms for Landsat 8 TIR bands. Surface skin temperature derived from Landsat 8 thermal images will be included in future versions of the database.*

SPECIFIC COMMENTS:

-Section 3.2, page 6, line 11: If I am not wrong the algorithm used by Simon et al 2014 is the same than the algorithm presented by Jimenez-Munoz & Sobrino 2009, so it is not a new version.
*That line would be more properly expressed as "The algorithm was implemented by Simon et al. (2014) and used for producing the LakeSST data set."*

-Section 5.3, page 13: Please include some comments about the SCL-off problem in Landast-7, and how this problem is addressed in the presented dataset. If Landsat-8 is not used for any technical resason (e.g. straylight problem), then some sentence could be also added in this section.
*On May 31, 2003 the Scan Line Corrector of the Landsat-7 satellite failed. As a result, the measurement scans cannot be corrected for the forward motion of the satellite and about 22% of an image data are lost. The gaps are less important in the centre of the image and increase towards the edges. Since this problem does not affect the radiometric and geometric corrections, SLC-off data could still be used for the creation of the dataset. No interpolation was applied to fill the data gaps. We recommend using the median SST provided in the dataset as an approximation of average lake skin surface temperature, since the median is not very sensitive to missing data and outliers.*

-Section 5.4, page 14, lines 31-32: "... we applied the algorithm also when clouds were partially present." This is a critical issue, because all the surface temperature retrieval algorithms working with Thermal-Infrared data are developed to be applied under clear sky conditions. This is well-known, so I think it has no sense to apply the algorithm in the presence of clouds. I would remove the data points contaminated by clouds and redo again the analysis to assess which are the main variables contributing to the seasonal bias. May be it is related to different atmospheric water vapor contents (?)

*Still, the current analysis allows to determine the limit of cloudiness under which the algorithm can be applied (approx. clearness index $k_t$ >0.6). When the data points contaminated by clouds are removed, a seasonal bias pattern is still present (see figure below). There is still a statistically significant relation between bias and solar radiation and air temperature. However, there is no statistically significant relation between atmospheric vapour content and bias.*

[Figure]

-Table 1: The header of the table should be more informative.

*We suggest replacing the previous heading by "Comparison of satellite-based temperature estimations made using the algorithm by Jiménez-Muñoz et al. (2009) ($T_{0.01\,m}$) to in situ measurements (temperature at 0.50 m, $T_{0.50\,m}$, and average temperature of the surface mixed layer, $T_{sml}$)."*

-References: in page 6 line 7-8 the authors refer to Sobrino 2004 with a strange symbol (#2738), but Sobrino 2004 is not included in the references list. Please correct.

*Ok.*

---

## Author Comment (AC2) · 9 Feb 2018

*Please find our response in italics.*

**RESPONSE TO ANONYMOUS REFEREE #2**

My recommendation is to publish the paper after some minor comments have been addressed:

Accuracy of 1.5 degrees is problematic. This is a very large discrepancy. Can you justify why this is sufficient for climate change studies?

*An RMSE for $T_{0.50m}$ of 1.51 ºC was calculated without any correction. If we remove the points affected by clouds as suggested by Reviewer #1 the RMSE decreases to 1.24 ºC. If we further use an average correction of $\Delta T$ = -0.41 ºC ($\Delta T_c$ = -0.46 ºC, according to COARE results, and $\Delta T_w$ = 0.05 ºC, according to measurements at Bimont), the RMSE decreases to 1.16 ºC. This is still a rather high uncertainty, but after the correction the satellite-based estimations of $T_{0.50m}$ are in average unbiased (mean absolute bias of 0.02 ºC).*

*A high uncertainty means longer time series may be necessary to determine trends (especially the feeblest ones). After deseasonalizing the 1999-2016 time series for the five validation sites, we calculated the linear trends in the table below (we indicate in bold the trends with p-value<0.10). Although the time series is relatively short (18 years), temperature trends may already be detected in two of the water bodies. If climate change is having an effect on the surface temperature of the other water bodies, longer data series may be necessary to detect this effect. The usefulness of the LakeSST for climate change studies will increase in the future as more data becomes available.*

| Lake code | Trend (ºC/yr) | p-value | n |
|-----------|---------------|---------|-----|
| AUL13 | -0.038 | 0.335 | 109 |
| BIM13 | -0.010 | 0.637 | 141 |
| LPC38 | -0.045 | 0.406 | 58 |
| PAV63 | **0.049** | **0.030** | 171 |
| SCR04 | **0.022** | **0.081** | 454 |

P2. L6: "4" should be "four". Any number less than 10 should be spelled out.

*Ok.*

Measurements with a thermal radiometer over the water bodies would be ideal. Then a more robust comparison of temperatures with radiometric temperatures is possible. Radiometric temperatures are a much better indicator of the performance of the surface temperature retrieval from satellite data.

*We totally agree with reviewer #2 but we do not have surface temperature data measured by radiometry yet. We are reflecting on how to obtain such data either by acquiring radiometric measurements through collaborations or by obtaining funding for the necessary instrumentation and field work.*

Include some comments about the SCL-off problem in Landast-7, and how this problem is addressed in the presented dataset. Discussion about why was Landsat 8 not included is necessary.

*On May 31, 2003 the Scan Line Corrector of the Landsat-7 satellite failed. As a result, the measurement scans cannot be corrected for the forward motion of the satellite and about 22% of an image data are lost. The gaps are lees important in the centre of the image and increase towards the edges. Since this problem does not affect the radiometric and geometric corrections, SLC-off data could still be used for the creation of the dataset. No interpolation was applied to fill the data gaps. We recommend using the median SST provided in the dataset as an approximation of average lake skin surface temperature, since the median is not very sensitive to missing data and outliers.*

*When this study was started, the issues with Landsat 8 TIR bands made us discard Landsat 8 data. Besides, the algorithm correcting stray light effects was only implemented into the Landsat processing system on February 2017. At the moment, some of the authors, in collaboration with other researchers, are testing different algorithms for Landsat 8 TIR bands. Surface skin temperature derived from Landsat 8 thermal images will be included in future versions of the database.*

I would highly recommend consultation with a native English speaker to clarify the language used. For example: P2. L13-15: this statement makes no sense.

*Thanks for the suggestion. We can replace the statement in P2, L13-15 by the following text:*

*"Research projects may also provide useful data. But in spite of the current efforts to make research data more accessible, it is still difficult to locate and access these data. In addition, the use of different instrumentation, methodologies and formats may require the homogenisation and formatting of the data sets prior to their use, requiring long treatment times."*

Much effort is given to the evaluation of various algorithms but more effort is needed to justify the use of daytime satellite retrievals and the impacts of the skin effect.

*Several practical reasons justify the use of Landsat thermal infrared images to study the surface temperature of inland water bodies:*

- *The use of satellite images allow the study of water bodies in a large spatial scale, even where field measurements are not possible or too costly.*

- *The high resolution of Landsat data allows the study of small and or narrow water bodies that could not be studied using other satellite data.*

- *The long period of availability of Landsat data allows the study of long term trends.*

- *Nighttime satellite TIR products do not offer the high resolution and long data availability that Landsat offers.*

*In addition, as mentioned above, we have decreased the RMSE of the surface bulk temperature estimations to 1.2 ºC after removing the data affected by clouds and correcting the cool skin effect. Further improvements could be made by correcting the warm layer effect. Since the performance of the COARE algorithm was not very good (coefficient of correlation of 0.45) we prefer not to use it to estimate the warm layer effect. Several one-dimensional hydrodynamic models are used to simulate water temperatures in the limnological community. However, their RMSE is typically of the same order of magnitude as the RMSE of the satellite-based estimations. In fact, because of the microclimatic conditions at inland water bodies, the use of an algorithm or model to correct the warm layer effect would require using in-situ (above the water) meteorological data that are rarely available.*

*At the reservoir of Bimont (BIM13), the average warm layer effect at 10 a.m. UTC (approximate time of satellite measurements) was 0.05 ± 0.21 ºC. This accounts for a small part of the total error. Of course, these are results for a single study site and might not be applicable to other sites. However, we do not have the data to make a similar analysis for other sites. We could include a comment in the paper explaining this.*

*At present, limnologists often work with temperature uncertainties of the same order of magnitude as those observed in satellite-derived data. One-dimensional hydrodynamic lake models often have uncertainties of 1-2 ºC. Temperature profiles are often measured at a single point in the water body, not taking into account the spatial variability of surface temperature. In fact, the average standard deviation of surface temperature in a given image was 0.3-0.5 ºC in the five validation sites (see table below).*

| Lake code | Average standard deviation (ºC) |
|-----------|--------------------------------|
| *AUL13* | *0.37* |
| *BIM13* | *0.33* |
| *LPC38* | *0.33* |
| *PAV63* | *0.34* |
| *SCR04* | *0.50* |

*For these reasons, we think the data set may be used to study the thermal behaviour of inland water bodies. What is important in that case is to take into account the uncertainty of the measurements, to which end the quality analysis included in the paper will be useful. Still, we would be happy to make other efforts to justify the use of the LakeSST if it is possible with the available data. Is it there any analysis you would recommend?*

Why include the numerous outlines of various french regions in Figure 1? These are not defined in the legend and only distract from the overall presentation of the figure.

*We have deleted the outlines of French regions.*

[Figure]

Fonts on the y-axis of Figure 4 are quite small and difficult to read.

*We have improved the figure.*

[Figure]

Figure 11 should include R2 and p values

*We made no regression on the Figure 11. The diagonal line represents the 1:1 line (information that will be included in the figure legend in the next version of the manuscript). So, we do not include p-values, but we do include the RMSE and $R^2$.*

[Figure]

**Figure 11: Comparison of satellite-based temperature estimations and *in situ* temperatures: a) temperature at 0.5 m, b) temperature of the surface mixed layer. The diagonal line indicates the 1:1 relation.**